# The Evolution of Plant Hormones: From Metabolic Byproducts to Regulatory Hubs

**DOI:** 10.3390/ijms26157190

**Published:** 2025-07-25

**Authors:** Jasmina Kurepa, Jan Smalle

**Affiliations:** Department of Plant & Soil Sciences, Martin-Gatton College of Agriculture, Food and Environment, University of Kentucky, Lexington, KY 40545, USA; jasmina.kurepa@uky.edu

**Keywords:** plant hormones, evolution, stress physiology, receptor signaling, modularity

## Abstract

As sessile organisms, plants adapt to environmental challenges through flexible developmental and physiological programs. Hormones play a central role in this adaptability, integrating environmental signals into coordinated responses that regulate growth and stress tolerance. Comparative studies across photosynthetic lineages reveal that several core hormone functions are remarkably conserved, despite major evolutionary changes in hormone perception, biosynthesis, metabolism, and transport. This conservation suggests that plant hormones have played a pivotal evolutionary role—not only preserving essential biological functions but also enabling increased complexity in plant form and function. A similar dual role is observed in evolutionary endocrinology in animals, where hormones contribute to the emergence and regulation of complex traits. We propose that hormones such as cytokinins, auxins, brassinosteroids, strigolactones, and abscisic acid originated as metabolic derivatives closely tied to core physiological functions essential for survival and reproduction, including reproductive success, nutrient sensing, and dehydration tolerance. Over time, these compounds were progressively integrated into increasingly sophisticated regulatory networks, where they now serve as central coordinators and key targets of evolutionary selection. This model advances our understanding of hormone evolution by providing a structured framework to interpret the persistence, specialization, and integration of plant hormones across evolutionary timescales.

## 1. Introduction

In evolutionary theory, natural selection acts across multiple levels of biological organization—from individual genes to complex polygenic networks that support integrated functional traits. These traits often involve trade-offs, particularly between immediate survival and long-term reproductive success. To navigate these trade-offs, many species have evolved mechanisms that temporarily suppress reproduction under stress, reallocating resources toward survival while preserving future reproductive potential. This flexibility is enabled by a modular organization of physiological and developmental systems, where distinct functions operate semi-independently yet remain tightly coordinated to maintain adaptive balance [1,2].

This raises a central evolutionary question: how are modular systems that govern complex traits selected for and maintained as organisms evolve greater structural and functional sophistication? Traits governed by these systems often depend on the coordinated activity of thousands of genes, placing sustained selection pressure on large portions of the genome. Understanding how natural selection shapes the stability and adaptability of these integrated networks is essential to explaining the origin and persistence of complex multicellular life.

A compelling insight comes from studies in animal endocrinology, where hormones function as systemic mediators that maintain coordinated traits across increasing levels of complexity [3]. Hormone regulatory networks—including biosynthesis, signaling, transport, and metabolism—enable evolutionarily flexible, system-wide coordination, allowing animals to adjust development and physiology in response to environmental change while preserving integrated trait function. In this context, hormones function as regulatory hubs within modular networks, supporting both the stability and adaptive refinement of essential biological processes during evolutionary transitions.

Plants also rely on hormones as central regulatory agents, but their sessile lifestyle has driven an evolutionary emphasis on developmental plasticity rather than behavioral mobility. Unable to escape unfavorable conditions, plants must adjust in place, dynamically modulating growth and physiology in response to environmental cues [4,5]. This plasticity is largely governed by plant hormones, which fine-tune tissue development, resource allocation, and stress responses. As a result, hormones are essential for ensuring that structures such as shoots and roots form appropriately under fluctuating conditions. The significance of this role is evident in hormone-insensitive mutants: those lacking responsiveness to major growth regulators—auxins (AUXs), cytokinins (CKs), gibberellic acids (GAs), strigolactones (SLs), and brassinosteroids (BRs)—can still form basic organs but often display severe dwarfism and poor environmental responsiveness [6,7,8,9,10]. Similarly, mutants insensitive to stress-responsive hormones like abscisic acid (ABA) or ethylene grow under optimal conditions but show hypersensitivity to a range of stresses [11,12,13,14].

Given their central regulatory functions, it is natural to ask whether plant hormones—like their animal counterparts—have played an evolutionary role in maintaining complex adaptive traits. Several key plant hormones—including AUX, CK, BR, SL, and ABA—exhibit functional conservation from algae to angiosperms [15,16,17,18,19,20,21,22,23,24,25,26]. This deep continuity is especially notable because many algae lack canonical hormone signaling components yet still produce and respond to these hormones [27,28,29].

Such patterns of conservation gain particular significance in the context of one of plant evolution’s most transformative events: the transition from aquatic to terrestrial life. Terrestrialization exposed early land plants to a range of novel challenges, including greater water and nutrient scarcity, increased ultraviolet B (UV-B) radiation, and the buildup of toxic metabolic waste products that could no longer be released into an aquatic environment [30]. The evolution of vascular tissues and upright growth introduced additional demands, such as structural support and light competition.

These pressures likely drove the diversification and refinement of hormone regulatory networks, enabling plants to preserve developmental flexibility and enhance stress resilience in increasingly complex terrestrial ecosystems. This pattern reflects modular evolution, whereby biological systems evolve through semi-independent functional units (modules) that can adapt or diversify without disrupting core physiological functions [31]. Under stabilizing selection, core processes remain conserved even as molecular components such as hormone biosynthesis, signaling, transport, and metabolism diversify to support novel environmental strategies [1,2]. Despite structural and mechanistic differences across lineages, plant hormones have remained tightly integrated within these regulatory modules, facilitating both functional stability and evolutionary innovation in environmental responsiveness.

In summary, plant hormones likely emerged from metabolic intermediates, evolving into core regulatory components of modular networks that balance growth, reproduction, and stress responses. This evolutionary trajectory—marked by functional conservation amidst molecular diversification—parallels hormone evolution in animals and highlights the role of modularity in the evolution of complex life. Framing plant hormones within this evolutionary developmental perspective enhances our understanding of plant adaptation and provides a predictive basis for bioengineering traits relevant to agriculture and environmental sustainability. This review develops that hypothesis and outlines potential pathways for the emergence, diversification, and functional specialization of plant hormonal systems across evolutionary timescales.

## 2. Plant Hormone Modularity

The evolutionary conservation of plant hormones across diverse lineages suggests that they are embedded in robust, adaptable regulatory frameworks. To understand how hormones contribute to plant fitness under shifting environmental conditions, it is useful to examine their organization in terms of functional modules—semi-independent systems that govern key physiological outcomes. Among these, the balance between reproduction and survival is one of the most critical.

In a recent review, we examined how plant hormones mediate the trade-off between reproduction and survival [19]. The following summarizes some of the main findings from that analysis. When considering how plant hormones mediate the balance between reproduction and survival, CK primarily promotes reproductive investment under optimal conditions. In contrast, other major hormones—such as AUX, ABA, and GA—function primarily to limit or redirect reproductive growth under unfavorable conditions, prioritizing short-term survival instead. This distinction positions CK as a specialized regulatory module that supports full reproductive output when environmental conditions permit, while the other hormone systems act to impose necessary restraints on reproduction under stress. CK also contributes to stress tolerance, particularly in shoots, reinforcing their association with reproduction. Unlike hormones such as AUX, SL, or ABA, which suppress shoot growth to conserve resources, CK enhances resistance to shoot-specific threats—such as pathogens and oxidative stress—that cannot be mitigated by growth reduction alone. By reinforcing the integrity of actively growing shoots, CK helps protect reproductive structures during vulnerable developmental phases. This dual functionality underscores CK’s strategic role in reproductive investment, while allowing for regulation by survival-focused pathways when necessary.

From an evolutionary perspective, it is reasonable to propose that a complex and fitness-defining trait like reproductive maximization would be governed by a single, dedicated hormonal module. Centralizing control in this way likely enhances robustness by minimizing signal conflict and ensuring consistent activation under favorable conditions. However, under adverse environmental conditions, the CK pathway is actively downregulated by hormone systems specialized in promoting survival—a regulatory shift that is well-documented [19,32,33,34,35,36,37,38] (Figure 1).

These well-being and survival hormones commonly operate through double negative signaling pathways, in which their activity is repressed under optimal conditions and released under stress in a quantitative way [19]. This design allows plants to respond rapidly and proportionally to changing environments without compromising reproductive growth when conditions are favorable. In contrast, CK signaling is built on a positive activation cascade, consistent with its role as the default driver of reproductive development [19].

Beyond their relationship to reproduction, it is also useful to classify well-being-focused hormones according to the core adaptive functions they regulate. Despite the diversity of habitats occupied by photosynthetic organisms, several physiological demands are nearly universal: acquiring water and nutrients and managing abiotic and biotic stresses. These demands are met by hormone-driven modules, each specializing in distinct but interdependent capabilities [39]. However, trade-offs among these modules are often unavoidable. For example, expanding the root system to increase water uptake may directly conflict with the need to halt growth altogether under severe drought. These trade-offs highlight the modular organization of hormone-regulated functions, enabling plants to flexibly prioritize competing physiological needs in response to changing environmental conditions.

The true sophistication of plant adaptation lies not only in modular specialization but also in the complex networking of these modules via cross-regulatory hormone interactions. Through these networks, one hormone can modulate the biosynthesis, signaling, or degradation of another, integrating diverse signals to coordinate growth, development, and stress responses [40]. Such interactions enable plants to prioritize physiological demands flexibly, avoiding rigid control hierarchies and instead relying on dynamic, context-dependent regulation. We further explore the implications of these networks in Section 4, where we examine how hormonal cross-regulation supports tiered responses to drought and other environmental stresses.

This integrated architecture raises important evolutionary considerations. The modular organization of hormone function—anchored by default reproductive promotion via CKs and conditionally activated stress responses via other hormones—provides a flexible yet coordinated framework for balancing survival and reproduction [19]. Understanding how this regulatory architecture originated and diversified is key to explaining how plants evolved such sophisticated control over complex adaptive traits.

## 3. A Model for the Evolutionary Emergence of Plant Hormone Systems

A potential answer emerges from examining the evolutionary history of CK across the photosynthetic lineage. CKs are found throughout the green world, including in algae and cyanobacteria, where they are consistently linked to the promotion of reproduction, a role tightly coupled with their stimulation of photosynthesis [19]. Notably, CKs are biochemical derivatives of nucleotide bases—fundamental components of DNA—and carotenoid precursors in the isoprenoid biosynthesis pathway, which are essential for photosynthesis [41]. This dual origin connects CK directly to two fundamental cellular processes: replication and carbon fixation. Because the shoot is the primary site of both photosynthesis and reproductive development, CK naturally became associated with reproductive success early in evolutionary history.

An evolutionary sequence can thus be envisioned [19]. Initially, CKs may have existed as metabolic byproducts—combined derivatives of nucleotides and isoprenoids—whose abundance correlated with the rates of DNA replication and photosynthesis, but without any dedicated hormonal function. These pre-CK molecules may have exerted weak, non-specific effects, perhaps by modulating enzyme activities allosterically or acting in a cofactor-like capacity. Indeed, recent studies have revealed that plants harbor many metabolites capable of such promiscuous biochemical interactions, which has been proposed as a plausible mechanism for the evolutionary emergence of hormones [42]. Nevertheless, testing this hypothesis remains difficult because there is no definitive way to confirm whether a metabolite functioned as a pre-hormone before it acquired a true hormonal role. Even if pre-hormone-like compounds and their biosynthetic genes are identified in algae, gene knockouts may yield no clear phenotype, as these metabolites likely exerted only weak, non-specific effects rather than acting through dedicated receptors or signaling pathways. Thus, the absence of a phenotype would not preclude their historical role as evolutionary precursors to hormones.

Over evolutionary time, mutations enhancing the capacity of pre-CK molecules to consistently promote reproduction and photosynthesis would have conferred a selective advantage, progressively transforming a passive metabolic correlation into an active, causative regulatory function. This would have gradually shifted the relationship from passive correlation to functional causation. As these regulatory connections became more established, the cytokinin module likely co-evolved with increasing organismal complexity—acquiring dedicated control through biosynthesis, metabolism, transport, and signaling pathways. This evolutionary trajectory parallels models proposed for animal hormones, where regulatory networks similarly expand in complexity to meet the demands of multicellular coordination and environmental responsiveness [3].

Based on this reasoning, we propose that the emergence and evolutionary progression of hormone modules include the following three phases (Figure 2):**Association Phase**—Hormones initially arose as metabolic derivatives of compounds that were stably correlated with key complex functions such as reproduction, nutrient management, water use efficiency, and stress defense.**Causation Phase**—Through mutation, these metabolic derivatives began to actively promote the functions they once merely reflected, shifting the relationship from correlation to causation and establishing their role as hormones.**Integration Phase**—As developmental and physiological complexity increased, these hormone modules acquired greater regulatory sophistication through the evolution of dedicated biosynthesis, metabolism, transport, and signaling mechanisms.

This conceptual framework offers a unified lens through which the evolutionary emergence of diverse plant hormone systems can be understood. We have previously demonstrated that this model also applies to AUX, which is derived from amino acids and closely linked to nutrient status [19]. A review of the literature suggests that this three-phase model also applies to BR (discussed further in Section 5), ABA (Section 6.1), SL (Section 6.2), and salicylic acid (SA) (Section 7). Each of these hormones is derived from compounds linked to essential physiological functions and has evolved to regulate specific adaptive traits.

## 4. Modular Hormonal Strategies for Drought Stress Adaptation

To further assess the utility of the three-phase model, we next examine how it maps onto specific hormone systems that coordinate drought responses—one of the critical selective pressures in plant evolution. If hormone systems evolved through the modular integration of core functions, we would expect to find distinct yet interacting hormone modules that coordinate stress adaptation across a gradient of drought severity. Drought provides an ideal case study because it demands more than a simple “on/off” response: It presents a continuum of environmental stress, with each level requiring qualitatively different physiological and developmental adjustments mediated by hormonal crosstalk.

To navigate this escalating challenge, plants have evolved not just a single stress response but a sequence of graduated, modular strategies. These adaptations correspond to increasing drought severity and are coordinated by specific hormonal modules that regulate the necessary physiological and developmental shifts. To meet these diverse challenges, plants deploy four broad adaptive strategies:**Avoidance**, through increased root growth and stomatal closure.**Tolerance**, which includes oxidative stress mitigation.**Escape**, via early flowering and senescence.**Quiescence**, such as dormancy of buds or seeds.

These strategies are governed by distinct hormone modules activated sequentially or in combination as drought severity increases. Importantly, drought responses rely on extensive hormonal crosstalk, ensuring that growth, stress responses, and reproductive timing are dynamically adjusted to the plant’s environmental context. For example, ABA and ethylene interact antagonistically to determine whether growth is suppressed or stress responses are prioritized [43]. Similarly, ABA modulates AUX transport, altering root architecture under water stress [44,45]. These interactions exemplify the complexity of hormonal integration required for drought adaptation. While this section focuses on drought, the same hormone networks are deployed in response to other stresses, such as nutrient scarcity and pathogen pressure. SLs, for example, modulate shoot branching and nutrient uptake under phosphate limitation [46,47].

While a complete molecular model of these cross-regulatory networks is beyond the scope of this review, it is well established that plant hormonal networks are inherently dynamic, continually reshaped by interactions among biosynthesis, signaling, transport, and feedback loops. Recent studies have mapped these networks, revealing key points of convergence and antagonism between hormone pathways under drought and other stress conditions [44,48,49]. Moreover, the hormones involved in drought adaptation—such as ABA, AUX, SL, and CK—are not limited to roles in water-use efficiency. Instead, they play central roles across a broad range of developmental and physiological processes, including root architecture, senescence, and reproductive timing [50]. Thus, their engagement in drought responses reflects the versatility and modularity of plant hormonal signaling systems rather than a specialization for stress adaptation alone.

### 4.1. CK: Default Growth Under Ample Water

Under well-watered conditions, CK signaling predominates [19]. CK promotes shoot growth and reproductive development while limiting root expansion to the minimal level necessary for support. Other hormone systems remain largely inactive, allowing maximal reproductive investment under optimal environmental conditions.

### 4.2. BR: Reproductive Persistence Under Mild Drought

When water becomes limited but not critically so, CK biosynthesis declines and BRs are upregulated [19]. BRs sustain shoot growth and reproduction but also activate adaptive features: They promote root growth and leaf senescence to improve water status and limit final shoot size. BRs also increase seed size, improving progeny survival under stress, though typically at the cost of reduced seed number. This shift marks the beginning of a trade-off strategy—still focused on reproduction, but with safeguards for survival.

### 4.3. AUX: Increased Water Acquisition and Conservation Under Moderate Stress

As drought stress increases, the AUX module becomes dominant (while CKs and BRs are repressed), shifting the plant to a more aggressive water acquisition and conservation mode [19,36,37]. AUX inhibits shoot growth and strongly promotes root expansion—particularly lateral root development—to enhance water uptake from upper soil layers. The plant still detects water in its accessible environment but invests strongly in uptake mechanisms, while reducing water loss via repression of shoot growth. Reproduction is further reduced, and vegetative growth is rebalanced in favor of survival. It needs to be noted here that, while the evolutionary emergence of AUX can be traced back to the ability to ensure optimal nutrient status in algae [19], this focus expanded during terrestrialization to include water status management, since the emergence of a root system—crucial for the survival of higher plants—was necessary for both nutrient and water uptake.

### 4.4. SL: Resource Prioritization Under Severe Drought

When water is scarce even in upper soil layers, the SL module is activated [51,52]. It promotes primary root elongation to access water located deeper in the soil, while strongly repressing lateral root formation, shoot branching, and photosynthesis predominantly via stomatal closure [53,54]. SL also promotes leaf senescence and early flowering, indicating a shift to a severe escape strategy. This module is expected to be triggered only if water is sensed in deeper soil layers, offering a final attempt to maintain some level of reproduction before full growth arrest becomes necessary.

### 4.5. ABA: Growth Arrest Under Extreme Drought

When water availability is critically low, the ABA module predominates [55,56]. It induces a global growth inhibition, including stomatal closure. While the other well-being hormones are also engaged in promoting stress tolerance involving reactive oxygen species (ROS) detoxification, this is strongly enhanced by the ABA module. ABA also triggers bud and seed dormancy, enabling survival through drought periods [55,56]. It has a dual effect on reproduction: under long-day conditions, ABA promotes flowering and senescence (an escape mode), whereas under short-day conditions, it inhibits flowering, placing the plant in a quiescent survival mode [19].

### 4.6. Conclusions

The modular architecture of drought-responsive hormone systems allows plants to implement mutually incompatible strategies—such as BR-driven shoot expansion versus SL- or ABA-induced growth suppression—based on the severity of environmental stress (Figure 3). Rather than relying on a single continuum of regulation, plants activate qualitatively distinct hormonal modules that engage in coordinated, often antagonistic, developmental programs.

Critically, the hormone modules are not only activated sequentially but are also functionally diverse and sometimes in direct opposition:CK and BRs differ in their effects on root growth, seed size, and senescence.BRs and AUX exert opposite influences on shoot growth and photosynthesis.AUX and SLs diverge in how they shape root architecture—promoting lateral versus primary root development, respectively.SLs and ABA differ in their balance between growth regulation and entry into dormancy.

Despite these contrasts, these hormones interact extensively—both within and across classes—enabling flexible, finely tuned responses rather than rigid, binary switches [57,58,59]. For example, AUX and SLs synergistically suppress shoot growth [19,60], while ABA and SLs coordinate stress signaling responses [61]. This flexibility ensures that plant development can be continuously adjusted to match fluctuating levels of water availability.

The sequential engagement of CK, BR, AUX, SL, and ABA also reflects a progressive withdrawal from reproduction as drought intensifies. BR marks the earliest phase of stress engagement, incurring only a mild cost to reproductive output. AUX and SLs increasingly reallocate resources away from reproduction toward survival. Finally, ABA imposes full reproductive arrest under extreme drought, often inducing quiescence through dormancy.

This tiered, modular organization not only equips plants to manage drought stress but also exemplifies how hormone evolution has tailored physiological responses to varying environmental challenges. We next examine this dynamic through the lens of specific hormone systems, starting with brassinosteroids.

## 5. Brassinosteroids (BRs)

BRs offer a compelling case for the three-phase model of hormone evolution. Among plant hormones, BRs share key functions with CKs, which we previously identified as a core module for reproductive maximization [19]. Like CKs, BRs promote shoot growth, enhance photosynthesis, and support reproductive development—traits associated with investment in progeny [19]. However, BRs differ from CK in a critical way: they also stimulate moderate root growth, suggesting a more balanced allocation of resources between shoot and root systems. This divergence reflects their distinct regulation—while CK biosynthesis is typically repressed under nutrient or water limitation, BR synthesis is induced under these same conditions. As such, BRs occupy a transitional role between default reproductive promotion and early stress adaptation, offering insight into how hormonal modules evolve functional diversity while remaining embedded in a conserved regulatory framework [19].

BRs also differ from CK in their roles during senescence. While CKs delay senescence to extend the shoot growth phase, BRs promote senescence, likely as an adaptive mechanism to limit the final size of the shoot as the main water-consuming organ [19]. This function allows BRs to sustain reproductive output when CK signaling is suppressed. Further supporting BRs’ role in reproduction under resource limitation is their ability to increase seed size—mainly by expanding endosperm volume—at the cost of reduced seed number [19]. This trade-off reflects a shift toward ensuring progeny survival rather than quantity, a hallmark of reproductive investment under constraint.

### 5.1. Association Phase: Biochemical Links to Reproduction and Dehydration Stress Tolerance

BRs are derived from sterols synthesized via the isoprenoid pathway, which also provides key precursors for photosynthesis [62,63]. As structural components of plasma membranes, sterols perform two critical functions: they enable rapid cell division by supporting membrane biogenesis, and they ensure membrane integrity and control water permeability to enhance cellular dehydration tolerance [64,65,66]. In addition to these structural roles, sterols are also involved as signals in the promotion of cell division and drought stress tolerance [64,67].

Because BRs are sterol derivatives, their synthesis is expected to rise during both vigorous growth and dehydration stress. This positions BRs at the biochemical intersection of reproductive development and drought stress resilience, making them strong candidates for hormone evolution through functional association.

### 5.2. Causation Phase: Early Functional Roles

BRs have been detected in various algae, where they appear to support photosynthesis, reproduction, and dehydration tolerance—functions they retain in land plants [17]. In addition, BR biosynthesis in algae is induced by osmotic stress, indicating that its function as a dehydration stress-induced reproduction promoter is ancient [17,18].

### 5.3. Integration Phase: Regulatory Sophistication in Land Plants

As plants adapted to terrestrial life, the BR module acquired greater regulatory complexity. In higher plants, BR perception involves the receptor BRI1 (BR-Insensitive 1), which activates downstream signaling by inactivating the repressor BIN2 (BR-Insensitive 2)—a classic double-negative regulatory mechanism [68]. This structure allows BR activity to be tightly modulated by environmental inputs, ensuring growth and reproduction are balanced with resilience.

The absence of the BRI1–BIN2 system in some early-diverging land plants suggests it evolved alongside the developmental and physiological advances of flowering plants [15]. This innovation allowed for more refined coordination between the BR and CK modules. Under optimal conditions, BR signaling is downregulated, allowing CK to drive full reproductive output. Under mild stress, BR activity rises as CK biosynthesis falls, enabling reproduction to continue under constrained conditions. This interplay illustrates the modular integration of hormonal signals to ensure fitness across diverse environments [19].

### 5.4. Conclusions

Together, these observations trace the full arc of BR evolution through the three-phase model: biochemical association with key functions, emergence of causal influence, and eventual integration into a sophisticated regulatory network. BR’s sterol-based origin links it to both proliferation and dehydration tolerance. Its apparent early regulatory roles in algae hint at an ancient shift from correlation to causation. Finally, its integration into land plant signaling via BRI1–BIN2 demonstrates how hormone modules evolve functional depth and coordination. In modern plants, BRs complement the CK module by sustaining reproductive investment when conditions begin to deteriorate—underscoring how hormonal evolution produces modular, cooperative systems that maintain adaptive balance in complex environments [19].

## 6. ABA and SL: Chloroplast-Derived Hormones for Severe Drought Stress

Among the five hormone modules ranked by their roles in dealing with the range of drought severity, ABA and SLs are upregulated under more extreme water-limiting conditions [61,69]. Both hormones are synthesized in the chloroplast and derived from carotenoids—ABA from xanthophylls, SL from carotenes—via the actions of carotenoid cleavage enzymes. While differing in specific developmental outcomes, ABA and SLs share overlapping roles in promoting plant survival during dehydration stress, with ABA largely responsible for the response to drought stress extremes [61,69].

### 6.1. ABA: An Ancestral Drought Tolerance Module

ABA is one of the most ancient hormone systems in the photosynthetic world and has consistently been linked to water-use efficiency [70]. Its emergence, diversification, and integration into increasingly complex signaling networks align closely with the proposed three-phase model of hormone evolution.

#### 6.1.1. Biochemical Origin: ABA as a Xanthophyll-Derived Metabolite

Chloroplasts are particularly vulnerable to drought-induced oxidative damage. The xanthophyll cycle, essential for photoprotection, dissipates excess excitation energy and suppresses ROS [71]. This cycle is activated under drought stress across all major photosynthetic lineages—including algae and cyanobacteria—indicating its ancestral role in oxidative stress defense [72].

ABA, a derivative of xanthophylls, is consistently synthesized in response to dehydration, making it a metabolite stably correlated with drought tolerance [70]. This aligns with the Association Phase of hormone evolution, in which metabolites linked to critical physiological traits later acquire regulatory functions.

#### 6.1.2. From Correlation to Causation: Early Functional Roles of ABA

The core drought-related functions of ABA—oxidative stress tolerance, growth inhibition, and dormancy induction—are conserved from cyanobacteria (the evolutionary origin of chloroplasts) to angiosperms [22,23,25,70]. ABA synthesis and action are induced in response to dehydration across this entire lineage. Although detailed signaling mechanisms in algae remain poorly understood, this long-standing functional conservation suggests an early evolutionary transition from a correlated metabolic marker to a causative regulator.

#### 6.1.3. Integration and Sophistication in Terrestrial Plants

As plants colonized land and faced increased dehydration risk, ABA evolved into a highly integrated and responsive regulatory module. This process peaked in angiosperms, which exhibit high water demands and require fine-tuned stress regulation. Key innovations that mark the Integration Phase include:Receptor evolution and double-negative signaling: The emergence of the PYR (Pyrabactin Resistance)/PYL (PYR1-Like) receptor family increased ABA sensitivity and, through its interaction with the protein phosphatase 2C (PP2C) class of ABA response repressors, enabled ABA signaling to remain quantitatively repressed under low-stress conditions and rapidly activated when needed [19,73]. A key evolutionary development in this receptor family was its enhanced ABA sensitivity and specificity in higher plants relative to primitive land plants, enabling more precise control through increased signaling amplitude [74,75].Enhanced stomatal control: Basal land plants show limited stomatal response, but angiosperms evolved dynamic ABA-mediated stomatal closure, enabled in part by advances in ABA transport [76,77].Dormancy regulation: Through innovations in ABA transport, this hormone also became a central regulator of seed dormancy, aligning germination and reproduction with seasonal moisture availability [76].

These features demonstrate how ABA co-evolved with increasing plant complexity to serve as a key adaptive module, integrating environmental sensing, developmental control, and survival under drought stress.

#### 6.1.4. Functional Versatility: ABA in Salt Stress Response

ABA’s role in salt stress further supports its adaptive versatility and how its linkage to dehydration stress survival is maintained. In salt-sensitive species (most higher plants), ABA suppresses root growth and promotes quiescence, conserving water. In contrast, in salt-tolerant species, ABA promotes root growth under salinity, thus enhancing water acquisition [78]. This context-dependent regulation underscores how ABA’s core evolutionary function—supporting dehydration stress tolerance—has been maintained and modulated across different environmental contexts. It also exemplifies how a conserved module can evolve plastic outputs while maintaining its fundamental adaptive role.

### 6.2. SL: A Parallel Chloroplast-Derived Drought Tolerance Module

SLs offer a compelling parallel to ABA as an ancestral hormone module linked to dehydration stress tolerance. Like ABA, SLs are derived from carotenoids and share key features of evolutionary origin and regulatory function, but with distinct physiological outcomes [61]. This section evaluates SL through the lens of the three-phase hormone evolution model.

#### 6.2.1. Association Phase: Biochemical Link to Dehydration Adaptation

SLs are synthesized from carotenoid precursors via the isoprenoid pathway—biochemical origins they share with ABA [61]. Carotenoid biosynthesis is upregulated under osmotic, salt, and drought stress across the green lineage, including algae, highlighting a conserved association between carotenoid metabolism and the dehydration response [79,80,81].

#### 6.2.2. Causation Phase: Functional Hormone Activity Before Land Colonization

Endogenous SLs were detected in some algal species and the key biosynthetic genes are present in algal genomes [82,83]. Functional relevance is further supported by experiments demonstrating that algae respond to exogenous SL application [20,82]. These findings suggest that a physiological response system capable of perceiving and reacting to SL signals existed prior to land colonization—supporting an early transition from metabolic association to hormonal causation.

#### 6.2.3. Integration Phase: Diversification of SL Function in Land Plants

With terrestrialization, the SL module acquired greater regulatory and functional complexity, becoming an integral part of the plant’s drought response network. Key evolutionary developments include:Adoption of double-negative signaling via D14 (Dwarf14) receptor and MAX2 (More Axillary Growth 2 F-box)-mediated degradation of repressors, allowing for conditional activation [53].Root architecture remodeling, where SLs promote primary root elongation while suppressing lateral roots—enhancing access to deeper water sources under drought [84].Coordination of escape strategies, including shoot growth suppression, stomatal closure, and induction of senescence [85,86].

SLs often act synergistically with ABA to coordinate these responses [61].

### 6.3. Summary

ABA and SLs provide strong validation for the proposed three-phase model of hormone evolution. Both hormones originated as chloroplast-synthesized metabolites closely linked to dehydration stress survival—fulfilling the Association Phase through their ancestral correlation with essential adaptive functions. Over time, they transitioned to causative regulators, as evidenced by their conserved roles in mediating drought responses across the green lineage, consistent with the Causation Phase. Their subsequent evolution into complex, multi-functional modules with specialized signaling pathways, regulatory feedback loops, and developmental integration exemplifies the Integration Phase (Figure 4).

Together, ABA and SLs demonstrate how metabolite-derived signals can evolve into distinct yet complementary hormonal systems that coordinate finely tuned responses to severe drought. While ABA governs tolerance and dormancy, SL drives architectural remodeling and escape responses. This dual system enables plants to respond adaptively across a range of extreme water-limiting conditions. Their evolutionary trajectories not only reinforce the biochemical and functional logic of the three-phase model but also underscore its explanatory power for understanding the emergence and diversification of hormone systems in plant evolution.

## 7. Salicylic Acid (SA): A Dual Role in Biotic and Abiotic Stress Responses

While the previously considered hormone modules (CK, AUX, BR, ABA, and SL) primarily emerged to manage specific physiological challenges—such as reproduction, nutrient acquisition, or water stress—SA stands apart as it integrates responses to both biotic stress (e.g., pathogen attack) and abiotic stress (notably UV-B radiation) [87,88]. The phenylpropanoid (PP) pathway, a central metabolic hub for stress resilience [89,90], provides the biochemical foundation for SA biosynthesis and points to an evolutionary link with general stress responses.

### 7.1. Association Phase: Biochemical Integration with Dual Stress Defense

SA synthesis is embedded in the PP pathway, which produces a diverse arsenal of defense-related compounds. The pathway begins with the conversion of phenylalanine into *trans*-cinnamic acid (*t*-CA), which is then converted into *p*-coumaric acid and subsequently branches into multiple biosynthetic routes [91]. One key branch involves the conversion of *t*-CA into benzoic acid (BA), which serves as the immediate precursor of SA [92].

This *t*-CA → BA → SA route is widely considered the ancestral biosynthetic pathway for SA in land plants, and was likely already established in algae, given that SA has been detected across diverse algal species [89,92,93,94]. Notably, both *t*-CA and BA themselves exhibit antimicrobial activity [95,96,97], and other PP-derived products contribute to cell wall modifications that inhibit pathogen spread and provide UV-B protection [90,98,99]. Thus, SA biosynthesis likely arose as part of a broader PP-induced stress response, establishing a biochemical correlation between SA levels and both biotic and abiotic stress conditions.

### 7.2. Causation Phase: Transition to a Regulatory Hormone

With the establishment of the PP pathway as a primary defense system [89], SA likely transitioned from a metabolic byproduct to an active regulator of plant immunity and abiotic stress responses. This shift may have occurred early in the green lineage, as SA has been shown to alleviate high light stress in algae [94]. The advent of novel terrestrial pathogens and intensified UV-B exposure likely exerted selection pressures that shaped SA’s functional evolution—favoring mutations that expanded its role in orchestrating systemic defense responses alongside enhancing mechanisms for UV-B tolerance and damage repair.

### 7.3. Integration Phase: Diversification of SA Signaling and Biosynthesis

The SA module has evolved a surprisingly complex regulatory framework in angiosperms. Central to this is the NPR (Nonexpressor of PR) family, which includes NPR1, an activator of SA-mediated defense, and NPR3 and NPR4, which act as SA response repressors [87]. These receptors modulate the activity of TGACG-BINDING FACTOR (TGA) transcription factors, orchestrating systemic acquired resistance. However, this NPR-based regulatory system is absent in basal land plants, even though SA still activates TGA-mediated immunity [87]. Furthermore, considerable evidence supports a regulatory, hormone-like role for SA in algae [94]. This suggests that early SA signaling operated through decentralized or alternative receptors, with NPR proteins representing a later evolutionary innovation that consolidated control.

Adding to this complexity, multiple SA-binding proteins have been identified, many of which control distinct subsets of SA-responsive genes [87,92]. This supports the idea of an initially modular sensing architecture, which evolved toward greater centralization over time.

Finally, SA biosynthesis also diversified. Although the *t*-CA/BA route is ancestral, the isochorismate pathway became prominent in later-evolving taxa, particularly in tissues specializing in immune defense [87,92]. This diversification supports the Integration Phase of the model, where multiple biosynthetic routes allow context-specific modulation of hormone levels.

### 7.4. Summary: SA and the Flexibility of Hormonal Evolution

SA aligns well with the proposed three-phase model of hormone evolution (Figure 5). In the Association Phase, SA emerged as a metabolite within the PP pathway, closely linked to both pathogen defense and UV stress mitigation. In the Causation Phase, it acquired the ability to actively initiate responses to these environmental challenges. During the Integration Phase, SA evolved complex receptor networks, context-sensitive signaling mechanisms, and multiple biosynthetic routes—hallmarks of advanced regulatory sophistication.

At the same time, SA presents an important extension to the model. Unlike other hormones that evolved around a single dominant function, SA’s dual role in biotic and abiotic stress responses suggests that multifunctional hormonal modules can arise early—especially when distinct environmental pressures converge. This dual functionality highlights the evolutionary flexibility of the hormone model and underscores how metabolic precursors can give rise to broad-spectrum regulators under conditions that favor integrated stress adaptation.

## 8. Other Plant Hormones

Unlike CK, BR, AUX, SL, ABA, and SA—which can be traced back to metabolites stably associated with foundational plant functions like reproduction, water and nutrient management, or stress tolerance—the evolutionary origins of GA, ethylene, and jasmonic acid (JA) remain unclear.

### 8.1. GA

The evolutionary origins of GA are beginning to be elucidated within the context of the three-phase model, although significant gaps remain. The Association Phase describes a hormone’s emergence from a metabolite consistently linked to an adaptive function. In the case of GA, the diterpenoid pathway produces intermediates such as *ent*-kaurene and *ent*-kaurenoic acid, which are present in many algal and early plant species [100,101]. While these compounds have not been directly linked to specific adaptive functions in algae, they may have conferred mild benefits related to membrane stabilization, redox homeostasis, or chloroplast protection—traits that would enhance fitness under fluctuating environmental stresses such as high light intensity, oxidative stress, or nutrient limitation [102]. Such baseline physiological benefits may have provided a foundation for their subsequent functional elaboration.

In the Causation Phase, the ecological demands of terrestrialization, particularly the need for vertical growth and shade avoidance, likely intensified the functional relevance of these diterpenoid intermediates. In bryophytes and basal land plants, growth responses to far-red light enrichment—resembling shade avoidance—are mediated not by bioactive GA but by their precursors [101]. This suggests that regulatory roles for diterpenoids predated the evolution of the complete GA biosynthetic and signaling pathway, serving as modulators of growth and light perception in early land plants.

The Integration Phase is marked by the development of specialized perception and signaling systems. In vascular plants, the GIBBERELLIN INSENSITIVE DWARF1 (GID1) receptor perceives bioactive GA and triggers the degradation of DELLA growth repressors, establishing a classical double-negative regulatory mechanism [8]. The absence of GID1 receptors in non-vascular plants, despite the presence of GA-like growth responses, underscores that the complete hormonal module evolved later, likely driven by competitive pressures for light acquisition in increasingly complex terrestrial ecosystems [100,101]. Thus, the evolutionary path of GA reflects a gradual co-option of primary metabolic intermediates into specialized regulatory networks, culminating in their pivotal role in coordinating growth and developmental plasticity in higher plants.

### 8.2. Ethylene and JA

The evolutionary origins of ethylene and JA remain uncertain, as current evidence does not establish clear ancestral biochemical links to foundational adaptive traits, unlike those documented for CK, BR, AUX, SL, ABA, and SA. A complicating factor is their unique physicochemical nature: ethylene is a gas, and JA is a volatile compound, both enabling interplant communication that may have influenced their evolutionary paths differently. The presence of JA in charophyte algae has been recently confirmed [102,103], whereas ethylene biosynthesis and perception are widespread across a broad range of organisms, including many non-photosynthetic lineages [104]. This distribution is likely due to convergent evolution, making it difficult to associate ethylene with a single ancestral adaptive function [105]. Additionally, ethylene’s role in modulating plant–microbiome interactions adds further complexity to understanding its evolutionary emergence [106]. Despite uncertainties about their origins, both ethylene and JA pathways have reached a highly integrated phase in higher plants. Ethylene is perceived by a subfamily of histidine kinases, referred to as Ethylene Histidine Kinases (EHKs), while CORONATINE INSENSITIVE1 (COI1) functions as the JA receptor [102,107].

## 9. Evolution Towards Receptor Dominance

The functional presence of key plant hormones—CK, AUX, BR, SL, ABA, and SA—has been widely documented in algae. In these early-diverging lineages, hormones contribute to reproduction, nutrient regulation, and stress responses. However, algae lack the canonical receptor-mediated signaling systems that characterize land plants, and there are also some significant differences between the hormone response pathways of higher plants and more primitive land plant species [15,54,74,75,92,94,108,109,110]. This discrepancy raises a central evolutionary question: if hormone-mediated functions existed before these specialized receptors, how and why did the current receptor-based systems evolve and achieve functional dominance?

### 9.1. Incorporation in the Evolutionary Model

Our three-phase model of hormone evolution includes (1) the Association Phase, where metabolite levels correlate with adaptive traits; (2) the Causation Phase, where metabolites begin actively promoting those traits; and (3) the Integration Phase, marked by the evolution of biosynthesis, metabolism, transport, and specialized receptor-mediated signaling [19,42]. Receptor evolution exemplifies this final phase, enabling high sensitivity, feedback regulation, and precise control through double-negative regulatory logic, especially for growth and survival-related hormones. Importantly, the shift to receptor dominance was not purely additive but substitutive, replacing earlier rudimentary perception systems.

### 9.2. Repurposing of Regulatory Networks

The emergence of higher plant receptors coincided with the repurposing and incorporation of pre-existing regulatory networks within evolving hormone modules [75,108,109]. Whole-genome duplication events (WGDs), such as those identified in the seed plant lineage [111], likely facilitated receptor diversification and the expansion of hormonal functions. This way, the regulatory impacts of hormone modules dramatically expanded, thus overtaking the previous more rudimentary versions. This is especially apparent in the terrestrialization shift, but also applies to the continuing evolution of hormone modules from those present in primitive land plants all the way to the angiosperms [15,54,74,75,92,94,108,109,110].

### 9.3. Exceptions

Receptor dominance is not uniform across hormone systems. SA signaling, for example, remains partially consolidated, involving multiple receptors such as NPR1/NPR3/NPR4 and diverse SA-binding proteins [87,92]. Alternative binding proteins that interact with CK, AUX, or ABA without mediating full responses persist in modern plants, acting as molecular fossils of ancestral signaling mechanisms [112,113,114]. Primitive land plants also retain distinct ABA pathways separate from the canonical PYR/PYL system, pathways that appear lost in higher plants [115]. These cases suggest that receptor consolidation was gradual, context-dependent, and shaped by lineage-specific pressures.

### 9.4. Evolutionary Nodes and Co-Evolution of Ligands and Receptors

The emergence of receptor dominance can be viewed as an evolutionary node—a point where novel receptor systems unlocked new regulatory potentials. WGDs [111] and gene duplications generated genetic redundancy, allowing neofunctionalization of receptors and diversification of signaling pathways. Simultaneously, ligands and receptors likely co-evolved, refining specificity and expanding the functional repertoire of hormones. These evolutionary nodes represent pivotal shifts that allowed hormones to regulate increasingly complex developmental processes, including tissue differentiation and organogenesis. This perspective is well-established in evolutionary biology and has been applied to plant innovations [116,117,118]. Notably, analogous patterns are seen in animal evolution, where the diversification of nuclear hormone receptors underpins vertebrate endocrine complexity [119].

## 10. Conclusions and Future Directions

This review synthesizes evidence for a three-phase model of hormone evolution. The model offers a unifying framework to explain the deep functional conservation of hormone-regulated core abilities—such as reproduction, nutrient and water management, and stress tolerance—across photosynthetic lineages from algae to angiosperms. By examining individual hormones through this lens, we show that many systems (e.g., CK, ABA, SL, SA) follow this evolutionary arc, while others (e.g., GA, JA, ethylene) reveal alternative or incomplete trajectories. The model helps clarify how hormones transitioned from metabolic byproducts into central regulatory hubs, and how increasing sophistication and complexity enabled the diversification of plant form and function.

To synthesize the current empirical support across hormone systems within our proposed three-phase model, we provide an overview summarizing the available evidence for each hormone at each phase of evolution (Table 1). The table offers a concise reference for future research directions, particularly for hormones such as GA, JA, and ethylene, where evolutionary trajectories remain incompletely understood. This synthesis underscores that while the model holds robustly for several hormones, others require deeper investigation, particularly regarding their early associations and causal roles in plant physiology.

Several open questions remain that point to important directions for future research. Addressing these questions will require coordinated efforts across evolutionary, physiological, and comparative research. Key priorities include identifying the early functional roles of pre-hormones in basal algae and cyanobacteria; elucidating noncanonical signaling mechanisms that operated before the emergence of dedicated receptors; exploring hormone-like compounds in non-plant lineages such as fungi and bacteria to assess whether their regulatory roles reflect convergence or deep homology; and tracing how ancestral receptor systems were lost or replaced by the high-sensitivity modules now dominant in land plants.

Future work should also integrate environmental context more explicitly, assessing how specific abiotic and biotic pressures shaped the trajectory of hormone evolution. Comparative studies between plant and animal hormone systems may further reveal convergent solutions to the challenges of coordinating complex multicellular life. Together, these investigations will deepen our understanding of hormone evolution and clarify how modular signaling architectures emerged to support the physiological complexity of modern plants.

## Figures and Tables

**Figure 1 ijms-26-07190-f001:**
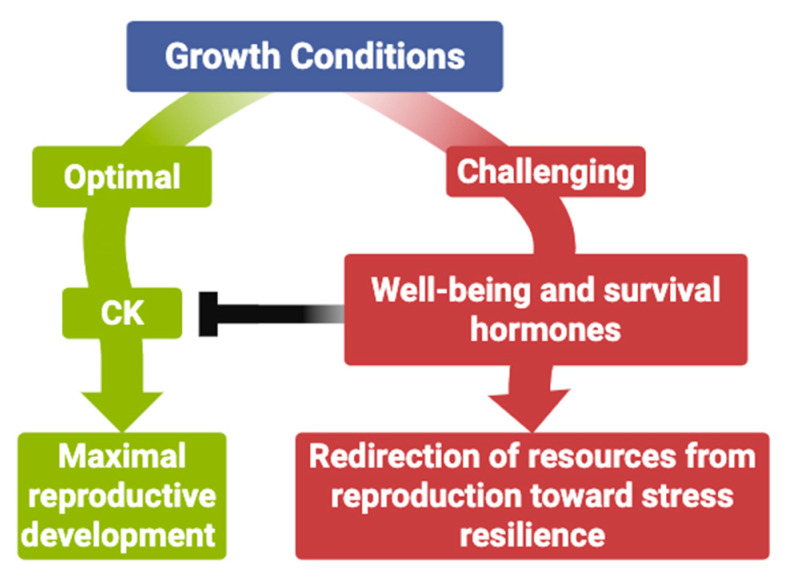
Cytokinins (CKs) are reproduction maximization hormones. CK promotes reproductive growth when conditions are favorable by enhancing shoot development and supporting resistance to shoot-specific stress (e.g., oxidative damage, pathogens). In contrast, survival-focused hormones such as auxins (AUX), abscisic acid (ABA), and gibberellic acids (GA) downregulate reproductive allocation under adverse conditions. CK functions as a permissive signal for maximal reproductive output and is repressed when environmental cues signal stress.

**Figure 2 ijms-26-07190-f002:**
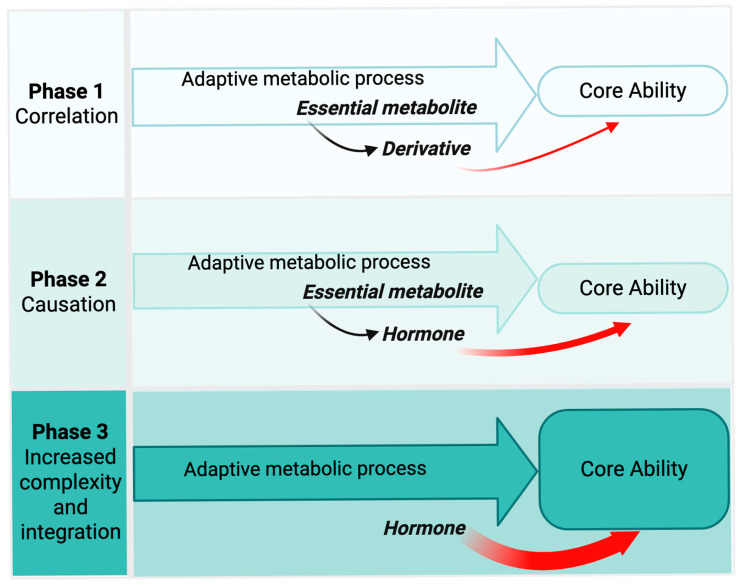
Model of hormone emergence and evolution. The model proposes three phases of hormone evolution: (1) Association Phase, where pre-hormone metabolites correlate with key adaptive traits such as reproduction or stress tolerance; (2) Causation Phase, in which these compounds acquire regulatory roles and begin actively promoting these traits; and (3) Integration Phase, marked by the evolution of hormone-specific biosynthesis, transport, metabolism, and high-sensitivity receptor-mediated signaling. The red arc indicates the strengthening regulatory role of the hormone across evolutionary time.

**Figure 3 ijms-26-07190-f003:**
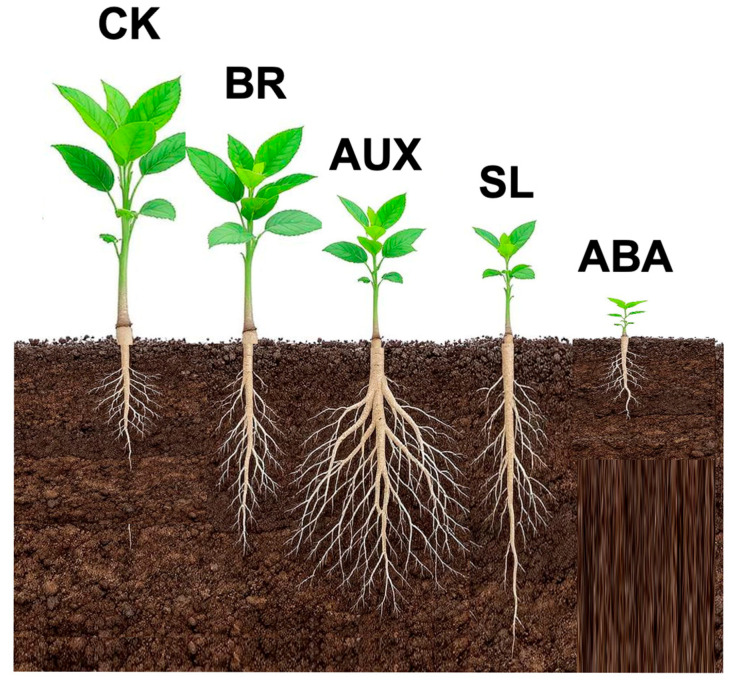
Modular hormone responses to progressive drought stress. This schematic illustrates the sequential activation of hormone modules as drought severity intensifies. Each module—CK, brassinosteroids (BR), AUX, strigolactones (SL), and ABA—initiates distinct physiological responses, enabling plants to progressively shift from maximal reproduction toward survival strategies such as root expansion, growth inhibition, and dormancy.

**Figure 4 ijms-26-07190-f004:**
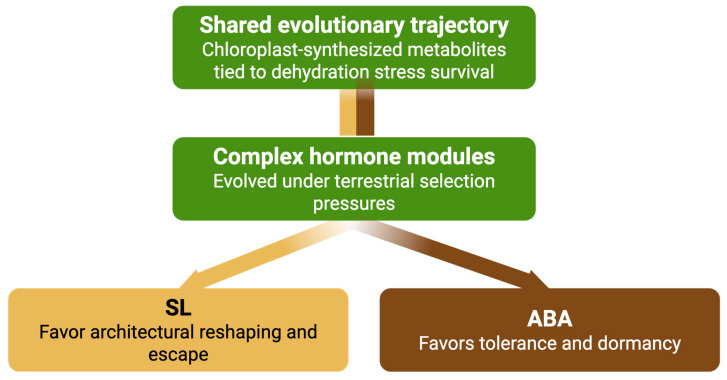
Divergent evolutionary trajectories of ABA and SL as chloroplast-derived hormone modules for dehydration stress adaptation. A conceptual model depicting the shared chloroplast-derived origins of ABA and SLs and their distinct evolutionary trajectories. ABA evolved as a module enhancing stress tolerance and dormancy, while SL diversified to promote architectural adjustments and escape responses under severe drought.

**Figure 5 ijms-26-07190-f005:**
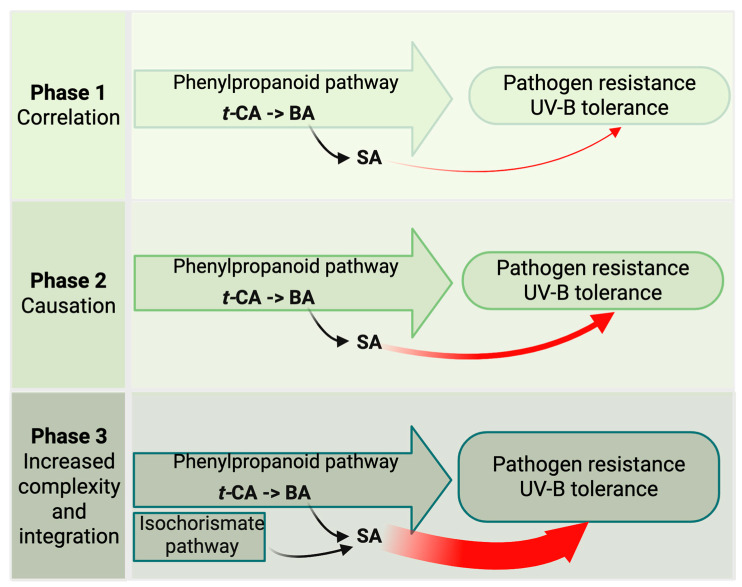
Emergence and diversification of salicylic acid (SA) as a hormone module. The figure maps the emergence of SA via the phenylpropanoid pathway, from *trans*-cinnamic acid (*t*-CA) through benzoic acid (BA), highlighting its ancestral link to stress defense and its subsequent regulatory specialization. The evolution of multiple SA receptors and biosynthetic routes reflects its dual role in biotic and abiotic stress adaptation.

**Table 1 ijms-26-07190-t001:** Summary of evidence for each hormone module across the three evolutionary phases of the proposed model.

Hormone	Association Phase ^1^	Causation Phase ^2^	Final Integration Phase ^3^
Cytokinins (CKs)	Nucleotides and isoprenoids	Reproduction and photosynthesis	CHK signaling
Auxins (AUXs)	Amino acids (tryptophan)	Nutrient management	TIR1/AFB signaling
Brassinosteroids (BRs)	Sterols	Reproduction, photosynthesis, and tolerance to mild drought stress	BRI1/BRL signaling
Strigolactones (SLs)	Carotenes	Dehydration tolerance	D14 signaling
Abscisic Acid (ABA)	Xanthophylls	Oxidative stress and dehydration tolerance	PYR/PYL signaling
Salicylic Acid (SA)	Phenylpropanoids	Pathogen resistance and UV-B tolerance	NPR signaling
Gibberellic Acids (GAs)	Unclear	Shade avoidance	GID1 signaling
Jasmonic Acid (JA)	Unclear	Wound response and inter-plant signaling	COI1 signaling
Ethylene	Unclear	Stress responses and inter-plant signaling	EHK signaling

^1^ The Association Phase column lists the key metabolites from which each hormone is derived. These metabolites contributed to essential physiological functions in ancestral organisms, preceding any regulatory role. ^2^ The Causation Phase column lists the core physiological functions that the hormone or its metabolic precursors began to actively promote. This marks their transition from incidental metabolites to functional signaling molecules. The table reflects the initial adaptive roles established at this stage, rather than the full diversity of functions these hormones control in higher plants. ^3^ The Final Integration Phase column lists the receptor-mediated signaling pathways through which each hormone is perceived in higher plants. The Final Integration Phase marks the emergence of dedicated receptor systems, accompanied by increased complexity in hormone biosynthesis, metabolism, transport, and signal transduction, enabling precise regulatory control over diverse physiological processes.

## Data Availability

This article is a review and does not involve the generation or analysis of new datasets. All data discussed are derived from previously published sources, which are appropriately cited within the manuscript.

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
