# Peer review of "The Evolution of Plant Hormones: From Metabolic Byproducts to Regulatory Hubs"

_ijms, 2025, doi:10.3390/ijms26157190_

Round 1
Reviewer 1 Report
Comments and Suggestions for Authors
This manuscript explores the hypothesis and outlines potential pathways for the emergence, diversification, and functional specialization of plant hormonal systems. There are several suggestions below to improve this manuscript:
- In the third section of the article, the author proposes a three-stage model based on extensive literature, but it lacks direct evidence. It is recommended to supplement with algal model experiments, such as knocking out CK precursor synthesis genes (e.g., IPT) in algae, to measure changes in photosynthetic rate and reproductive capacity, thereby validating the necessity of the association between metabolites and function.
- In the fourth section, the author presents a case study on drought, but only briefly discusses the cross-regulation and interaction of hormones without providing molecular-level evidence to clarify the mechanism of cross-regulation. It is suggested to add a cross-regulation model or cite studies on hormone interactions.
- In sections 5–9, the author overly emphasizes the evolutionary pressure of hormones on terrestrial adaptation while neglecting other pressures affecting hormones. For example, SL (strigolactone) is not only influenced by water stress but also related to nutrient acquisition. It is recommended to refine the discussion by categorizing different hormones and their respective pressure sources.
- In section 10, the author mentions that many receptor systems are absent in lower plants but present in higher plants, yet fails to provide the species distribution of gene loss or clarify the association between receptor family evolution and functional complexity. It is suggested to supplement with a phylogenetic tree of receptor family evolution and cite genomic and transcriptomic data.
- From an overall perspective, the article primarily highlights the functions of individual hormones (e.g., CK regulating reproduction, ABA regulating dormancy) but does not discuss the interconnections within multi-hormone regulatory networks. It is recommended to add a model and discussion on hormone coordination networks.
- The evolutionary model proposed by the author in the article is described as a gradual process, without explaining the abrupt changes at key nodes. For example, hormones in algae only regulate cell division, but how did they develop organ differentiation functions in land plants? It is suggested to add an analysis of evolutionary nodes in the hormone model.
Reviewer 2 Report
Comments and Suggestions for Authors
In this manuscript, Jasmina Kurepa and Jan A Smalle proposed that hormones such as cytokinins, auxins, brassinosteroids, strigolactones, and abscisic acid originated as metabolic derivatives closely tied to core physiological functions essential for survival and reproduction, including reproductive success, nutrient sensing, and dehydration tolerance. I have following comments:
1, Correspondence of authors should be provided in the revise manuscript.
2, The present title is convoluted, please modify it into a concise title.
3, For the introduction, main conclusions and practical interests of this study should be stated in the last paragraph of this section.
4, Figures 1, 3 and 4 are very gratuitous and ambiguous. The presentation of this figure is also very ambiguous
5, Many references are lacking. For instance, there are no citation in the paragraph 2.
6, A figure depicting origin and evolution of all phytohormones discussed in this manuscript is appealing.
7, Authors should consider to discuss the evolution of hormone biosynthesis and signaling separately.
Reviewer 3 Report
Comments and Suggestions for Authors
This review article synthesizes comparative, evolutionary and physiological evidence to propose a three-phase model (Association → Causation → Integration) for the emergence of major plant-hormone systems. After presenting the conceptual framework, the authors examine each hormone group (CK, BR, AUX, SL, ABA, SA, GA, ethylene, JA) in turn, then discuss receptor-driven signal evolution and open research questions. The narrative is ambitious and broadly coherent; figures summarize the modular drought-response cascade and evolutionary trajectories effectively and brings together disparate molecular, evolutionary and ecological data into a unifying model; breadth will interest both molecular biologists and evolutionary ecologists. My comments are appended below for consideration:
- Inreased” (l.248) and multiple duplicated citation blocks; some figure captions repeat surrounding text almost verbatim. I suggest careful copy-editing to remove typos and streamline captions.
- Hormones such as GA, JA and ethylene are labelled “unclear” yet still discussed at length; empirical gaps could be sign-posted more concisely. I would suggest Shortening speculative sections or add meta-analysis/data-table showing evidence density per hormone.
- Figures 1–4 are conceptually helpful but currently appear as low-resolution raster images within the PDF.
- Referencing is generally thorough, but several in-text numbers (e.g., “[19]”) are reused for very different claims, suggesting reference-manager glitches. A cross-check of the bibliography against in-text calls is advised.
With minor textual polishing, figure enhancement and clearer discussion of evidence gaps, this review will make a valuable contribution to plant-hormone evolutionary biology. The declared, limited use of AI tools for image generation and language critique is transparent and does not undermine scholarly integrity.
Round 2
Reviewer 1 Report
Comments and Suggestions for Authors
It could be accepted in the current version.
Reviewer 2 Report
Comments and Suggestions for Authors
Authors have carefully revised this manuscript according to my suggestions.